# Experimental Study and Theoretical Analysis of Steel–Concrete Composite Box Girder Bending Moment–Curvature Restoring Force

Jingjing Qi [1], Yining Ye [1], Zhi Huang [1,*], Weirong Lv [1], Wangbao Zhou [2], Fucai Liu [3] and Jidong Wu [1]

[1] School of Civil Engineering, Hunan University of Science and Technology, Xiangtan 411201, China
[2] Hunan Architectural Design Institute Limited Company, Changsha 410012, China
[3] Guangdong Gaiteqi New Materials Technology Co., Ltd., Qingyuan 511600, China
* Correspondence: huangzhi@hnust.edu.cn

**Abstract:** A steel–concrete composite box girder has good anti-seismic energy dissipation capacity, absorbs seismic energy, and reduces seismic action. It is very suitable for high-rise and super high-rise mega composite structure systems, which is in accordance with the condition of capital construction. In order to accurately study the elastic–plastic seismic response of the composite structure, the restoring force model of the building structure is the primary problem that needs to be solved. Previous research shows that shear connection degree, force ratio, and web height–thickness ratio are the major factors that influence composite box girder bearing capacity and seismic behavior. In this paper, low cycle vertical load tests of four steel–concrete composite box girders were conducted with different shear connection degrees and ratios of web height to thickness. The seismic behavior of a steel–concrete composite box girder was analyzed in depth, such as the hysteresis law, skeleton curve, and stiffness degradation law, etc. The influence of the shear connection degree and ratio of web height to thickness on seismic performance of the steel–concrete composite box girder was investigated. A three-fold line model of the bending moment–curvature skeleton curve of composite box girders was established. On the basis of experimental data and theoretical analysis, the formula of positive and negative stiffness degradation of composite box girders was obtained. Furthermore, the maximum point orientation hysteresis model of the bending moment–curvature of steel–concrete composite box girders was established. The calculated results of the restoring force model agree well with the experimental results. The accuracy of the proposed method is verified. The calculation method of the model is simple and clear, convenient for hand calculation, and suitable for engineering applications.

**Keywords:** steel–concrete composite box girder; restoring force model; shear connection degree; skeleton curve; bending moment–curvature; hysteresis rule





## 1. Introduction

A steel–concrete composite box girder is composed of a concrete slab and a grooved steel girder, and the concrete slab and grooved steel girder are connected by shear keys, so that the steel beam and concrete slab bear a common force. The steel–concrete composite box girder not only has the advantages of a light deadweight and larger torsional rigidity, which adapts to modern construction, but also gives full play to the merits of steel and concrete material and makes full use of the compression capacity of concrete and the tensile capacity of steel. Thus, it is widely used in bridges and large span buildings, which is in line with the future development direction of architecture [1,2].

In recent years, earthquakes occur frequently all over the word, and more and more attention has been paid to the seismic capability or seismic performance of structures. Daniels et al. [3] in 1970 took the lead in studying the seismic performance of composite box girders in a composite frame system. The elastic–plastic analysis method of composite box girders was obtained. Humair [4], Gowda [5], and Taplin [6] conducted tests on the seismic

performance of steel–concrete composite beams under low cyclic loading, successively. The influence of parameters of high strength steel and steel on mechanical properties such as load–displacement hysteresis curves of steel–concrete composite beams and the growth rate in relation to the shear connection degree-slip were discussed. The test results showed that the steel and concrete composite beams have good ductility performance. The increase in the rigidity of the web can improve the seismic performance of composite beams. China is a multi-earthquake country, and there have been many disastrous earthquakes in history. Therefore, China also pays more attention to the seismic performance of composite beams. Nie, J.G., etc. [7–9], studied the deformation and energy dissipation of steel–concrete composite beams under low cyclic loading. Considering the influence of the shear connection degree, the restoring force model of composite steel–concrete beams was established. The above research shows that steel–concrete composite beams have good anti-seismic energy dissipation capacity, absorb seismic energy, and reduce seismic action. It is very suitable for high-rise and super high-rise mega composite structure systems, which is in accordance with the condition of capital construction. However, the current research on the seismic performance of a composite beam mainly focuses on the composite I-beam, and the research on the resistance theory of composite box girder members lags behind the practical engineering application of composite box girders. It is of great significance to carry out seismic research on composite box girder members and to rationally analyze and evaluate the seismic performance of composite box girders to improve the theory and method of the seismic design of composite box girders, to improve the ability of major engineering structures to resist earthquake disasters, and to promote the application of composite box girders in mega-structures and long-span structures.

The seismic code stipulates that it is necessary to perform elastoplastic analysis of structures when they are subjected to rare earthquakes. In order to accurately study the elastic–plastic seismic response of the structure, the restoring force model of the building structure is the primary problem that needs to be solved [10–12]. Ashraf, etc. [13], proposed an elastic–plastic composite beam element which considered composite effect. Based on the elastic–plastic finite element unit, the composite beam analysis program was developed. The seismic performance of composite beams was calculated using this program, and the results were compared with the experimental results. The effectiveness of the composite beam elastoplastic unit was proven. Jiang, L.Z., Xin, X.Z., etc. [14–16], conducted vertical low cyclic loading testing of steel–concrete composite beams and box girders. The skeleton curve models, and the hysteresis rules of the continuous composite girder and simple support composite girder were proposed, and the load–mid-span deflection restoring force model of the continuous composite girder was established. The above studies on the restoring force model mainly focus on the composite I-beam and do not consider the influence of the composite interface slip on the restoring force model.

Gattaca [17,18] and Taplin [19] conducted experimental research on the seismic behavior of shear connectors under the action of repeated cyclic loading, successively. The load–deformation hysteresis curve of the shear connector was obtained and agreed well with the theoretical calculation results. Liu, J. [20] established a 3D FE model and performed a pseudo-static analysis of steel–concrete composite beams. Ding, F.X. [21] investigated the seismic behavior of a simply supported steel–concrete composite I-beam and box-beam through a quasi-static experimental study. A total of 22 composite beams were included in the experiments, and parameters including shear connection degree, transverse reinforcement ratio, longitudinal reinforcement ratio, section type, diameter of stud, and web thickness were investigated. From the above research, results showed that shear connection degree, force ratio, and steel girder width–thickness ratio are the major factors that influence bearing capacity and seismic behavior. Transverse reinforcement, section form, and stud diameter are the secondary factors. Thus, the influence of parameters such as shear connection degree and web height–thickness ratio cannot be ignored when constructing the restoring force model.

So far, most of the experimental and analytical study of composite box girders focus on its static performance and fatigue life [4]. Furthermore, research on the seismic performance of composite box girders is rare. Based on the low cyclic loading test and theoretical analysis of four steel–concrete composite box girders [22], the moment–curvature trifold line skeleton curve model and the degenerate trifold line maximum point orientation hysteresis model were proposed. The calculation curve of the model coincides well with the test curve. The proposed model calculation method is simple and easy to calculate, which is suitable for engineering applications.

## 2. Summary of Test

### 2.1. Specimen Design and Production

The hysteresis performance test of four steel–concrete composite box girders was carried out with the main parameters of high-thickness ratio and shear strength. The detailed information of the specimen is shown in Table 1, the section parameters are shown in Figure 1, and the longitudinal section is shown in Figure 2, where $r$ is the shear connection degree, $r = \frac{n_r}{n_f}$, $n_r$ is the actual number of studs in shear span, $n_f$ is the number of studs required for full shear connection, $\frac{b_f}{t_f}$ is the height–thickness ratio of the steel girder web, $d$ is the diameter of the stud, and $l$ is the spacing of studs. The thickness of the concrete cover was 15 mm. The steel beam plate was set with a longitudinal stiffener with a height of 50 mm and the stiffening rib thickness was 10 mm. According to DL/T 5085-1999 [23] and the existing literature research results [1,2,14], when the shear connection degree is greater than 1.00, its effect on the seismic performance of the composite box girder is very small. The range of the shear connection degree in this test was 0.44~0.71.

**Table 1.** Summary of steel–concrete composite box girder test specimens (mm).

| Serial Number | SCB-1 | SCB-2 | SCB-3 | SCB-4 |
|---|---|---|---|---|
| Span | 3000 | 3000 | 3000 | 3000 |
| $b_c$ | 650 | 650 | 650 | 650 |
| $t_c$ | 60 | 60 | 60 | 60 |
| $b_y \times t_y$ | 60 × 9.42 | 60 × 9.42 | 60 × 9.42 | 60 × 9.42 |
| $b_b \times t_b$ | 280 × 9.42 | 280 × 9.42 | 280 × 9.42 | 280 × 9.42 |
| $b_f \times t_f$ | 115 × 7.22 | 117 × 7.22 | 116 × 3.36 | 162 × 7.22 |
| $d$ | 12.8 | 12.8 | 12.8 | 12.8 |
| $l$ | 130 | 90 | 90 | 90 |
| $\frac{b_f}{t_f}$ | 15.9 | 16.2 | 34.5 | 22.4 |
| $r$ | 0.44 | 0.71 | 0.66 | 0.64 |
| Stirrup | 22Φ6 | 22Φ6 | 22Φ6 | 22Φ6 |
| Longitudinal bar | 13Φ13.4 | 13Φ13.4 | 13Φ13.4 | 13Φ13.4 |

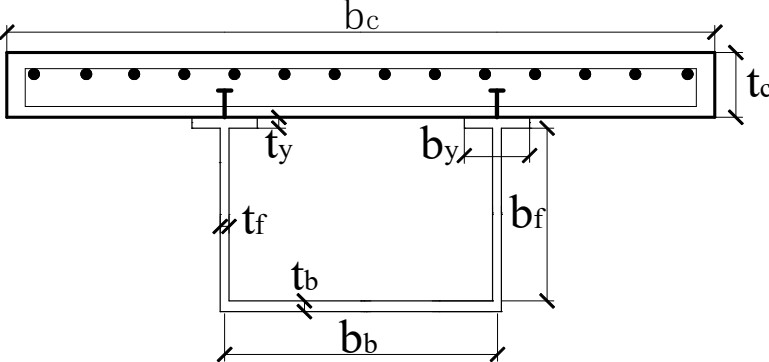

**Figure 1.** Section of composite box girder.

**Figure 2.** Longitudinal profile of composite box girder specimen.

The steel box girder was welded by a steel plate, and the steel girder flange studs were welded by the special bolted welder. The C40 concrete was used for its concrete slab. In the pouring process, the vibration rod was adopted to vibrate the concrete. The concrete slab was covered by cotton batting to keep moisture curing for one week, then it was maintained in the natural state for three weeks, and the curing time was a total of 28 days. In the same period, three groups (3 of each group) of 150 mm × 150 mm × 150 mm cube concrete compression test blocks were produced with the same maintenance condition as the composite box girder. The mechanical performance tests of the cube concrete compression block and composite box girder were carried out during the same period. The concrete elastic modulus was $E_c = 35,765$ Mpa, the axial compressive strength was $f_c = 46.56$ Mpa, the axial tensile strength was $f_t = 3.90$ Mpa, and the cube compressive strength was $f_{cu} = 58.20$ Mpa. The strength of the steel plate was determined by a tensile test, and the steel plate was tested with a standard specimen. The mechanical properties of the studs were provided by the supplier. The mechanical properties of the steel plates and bolts are shown in Table 2.

**Table 2.** Mechanical properties of steel (Mpa).

| Steel Category | Yield Strength $f_y$ | Ultimate Strength $f_u$ | Modulus of Elasticity $E_s$ |
|---|---|---|---|
| 4 mm Steel plate | 369 | 465 | 206,000 |
| 8 mm Steel plate | 273 | 400 | 200,000 |
| 10 mm Steel plate | 301 | 420 | 209,000 |
| Φ14 reinforced | 459 | 560 | 206,000 |
| Φ6 reinforced | 550 | 680 | 200,000 |
| Φ13 stud | 350 | 435 | 206,000 |

## 2.2. Test Device

The physical diagram and schematic diagram of the composite box girder loading device are shown in Figures 3 and 4. The composite box girder was loaded at two points in one-third and two-thirds with the distribution beam.

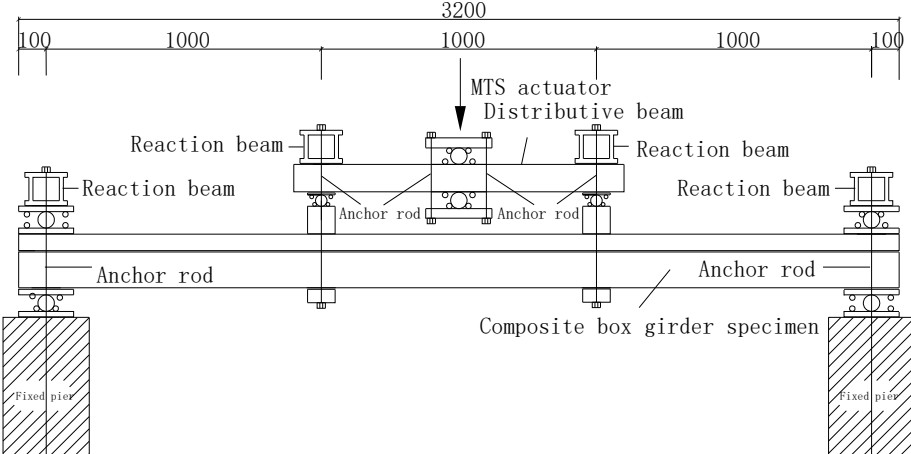

**Figure 3.** Schematic view of loading device of composite box girder.

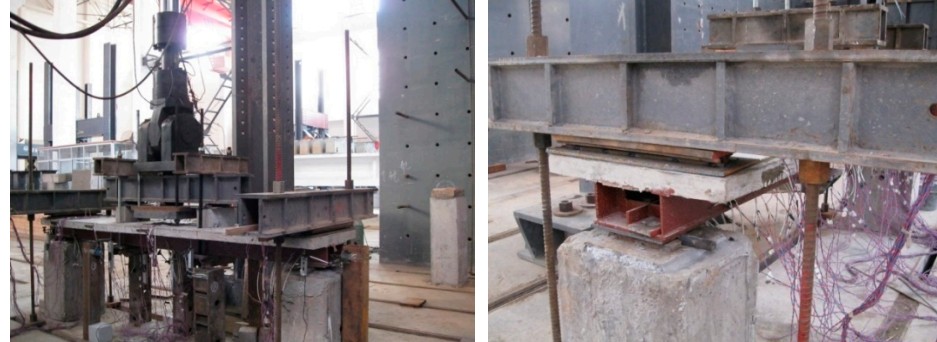

**Figure 4.** Physical view of loading device of composite box beam.

The distribution beam was connected with an MTS hydraulic servo actuator which was fixed in the vertical direction of the reaction frame and used to apply vertical cyclic load or displacement. The distributive beam ends were connected with the composite box girder using the anchor bolt, so that the distribution beam could exert upward force and downward force. The composite box girder was mounted on the reinforced concrete abutment, and the connecting piece was connected with the laboratory geosyncline by the anchor bolts, so that the beam end could bear the upward and downward support reaction.

### 2.3. Loading System

According to JGJ101-96 Specification of Testing Methods for Earthquake Resistant Building [24], the loading system of the specimen was developed, and the force–displacement hybrid control method was used to load the specimens in a low cycle. The low cyclic loading system is shown in Figure 5. The specific loading steps were:

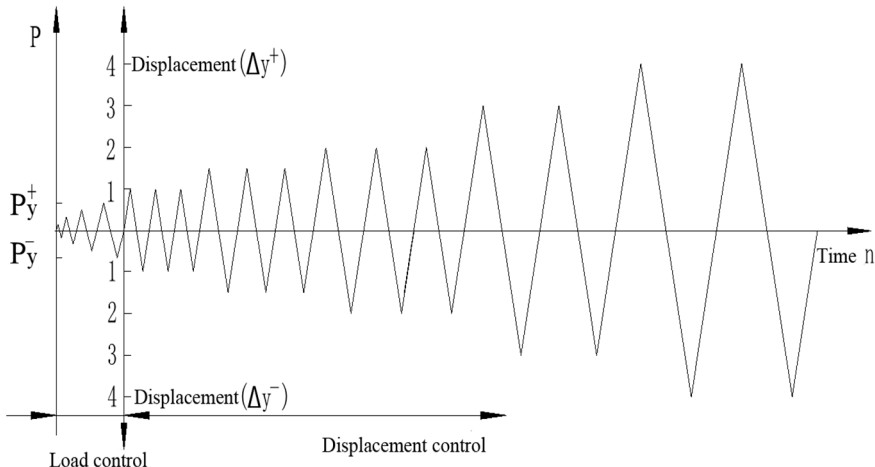

**Figure 5.** Loading process of low frequency cyclic loads.

Preloading: The composite box girder specimen was preloaded for positive and negative, respectively, with a 0.25 times negative concrete cracking load, eliminating the influence of initial defects in the device and checking whether all the instruments and devices were working properly.

The loading process for both positive and negative was divided into four equal steps until a negative concrete cracking load occurred. Then, the girder was unloaded at the constant speed of four equal steps. It was the first cycle.

The three-stage loading process with an incremental load control was adopted for the positive until reaching the steel beam yielding load $P_y^+$. The same method was used for the negative until reaching the yielding load $P_y^-$ of reinforcement in the concrete slab. The

positive and negative yield displacement was $\Delta_y^+$ and $\Delta_y^-$, respectively. The load at each stage was divided into four equal steps with constant loading and unloading as a cycle.

The loading method was shifted to the displacement control method. The positive and negative loading were respectively applied according to their respective yield displacement multiples, namely, $\Delta_y$, $1.5\Delta_y$, $2\Delta_y$, $3\Delta_y$, $4\Delta_y$, ..., $n\Delta_y$. The first three levels of load were evenly divided into five steps and cycled three times with a constant speed of loading and unloading. The latter load was evenly divided into five steps with a constant speed of a loading and unloading cycle twice.

### 2.4. Test Content

The two points of the composite box girder at 1/3 and 2/3 were loaded by means of the distribution beam. The main contents of the test were the positive and negative vertical deflections of the girder at $1/2$, 1/3, 2/3, and beam end support, concrete slab strain, reinforcement strain, steel girder strain, and stud strain, etc. The layout of the vertical deflection measurement points is shown in Figure 6.

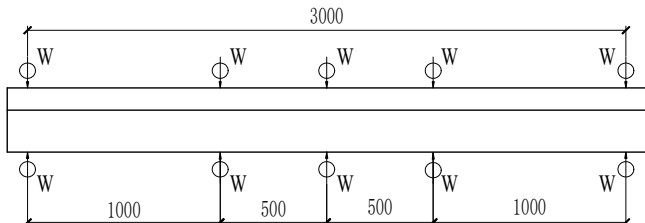

**Figure 6.** Measuring-point arrangement of vertical displacement.

## 3. Test Results and Analysis

### 3.1. Failure Mode

Under low cyclic loads, failure modes of composite box girder specimens can be classified into four categories due to the difference of the shear connection degree and web height–thickness ratio (Figure 7). The longitudinal shear failure occurred in specimen SCB–1, which was characterized by bond lifting between the concrete slab and steel beam and longitudinal splitting of the concrete slab in the flexural shear zone. Specimen SCB–2 showed flexural shear failure, which was characterized by the crushing of the concrete slab at the beam end, the longitudinal splitting of the concrete slab in the flexural shear zone, and the crushing of the flange of the concrete slab. Specimen SCB–3 showed local buckling failure, which was characterized by local buckling of the web, bond lifting between the concrete slab and the steel beam in the flexural shear zone, the crushing of the concrete slab at the beam end, and the longitudinal splitting of the concrete slab. Specimen SCB–4 showed bending failure, with the concrete slab at the beam end and the concrete slab flange in the flexural shear area being crushed.

### 3.2. Load–Deflection Hysteresis Curve

The mid-span displacement of the specimen was obtained by subtracting the measured settlement of the support from the measured mid-span displacement. The bending moment–curvature hysteresis curve of four composite box girder specimens is shown in Figure 8. The bending moment–curvature hysteresis loop of four composite box girders is relatively plump without an obvious pinch phenomenon.

(1) Before cracking, there is a linear relationship between the bending moment and the curvature, and the specimen is in the elasticity phase. The mid-span deflection is very small, and there is no residual deformation after unloading.

(2) After cracking, the bending moment and curvature hysteresis loops begin to show the shape of the curve. With the increase in the bending moment, the hysteretic curve begins to tilt significantly toward the deflection axis, and it presents obvious elastic–plastic properties. After unloading, the residual deformation gradually increases, and the hys-

teresis loops tend to be plump. The area of the hysteretic curve increases gradually, and the loading and unloading stiffness of the specimens gradually degenerate. However, the stiffness degradation is not significant, which shows good seismic performance.

(3) After the load exceeds the limit load, part of the steel yield and the yield strain of the steel increased. Loading and unloading stiffness of the specimens further reduce, and the unloading stiffness remains approximately elastic, while residual deformation increases further. An obvious approximate platform segment appeared in the curve, and the displacement increased continuously. After the approximate platform section, the displacement increases further, while the load decreases continuously, resulting in the final failure. The four composite box girders all have long approximate platform segments under positive loading, and short approximate platform segments under negative loading, indicating that the positive deformation capacity of the composite box girder is much stronger than the negative deformation capacity.

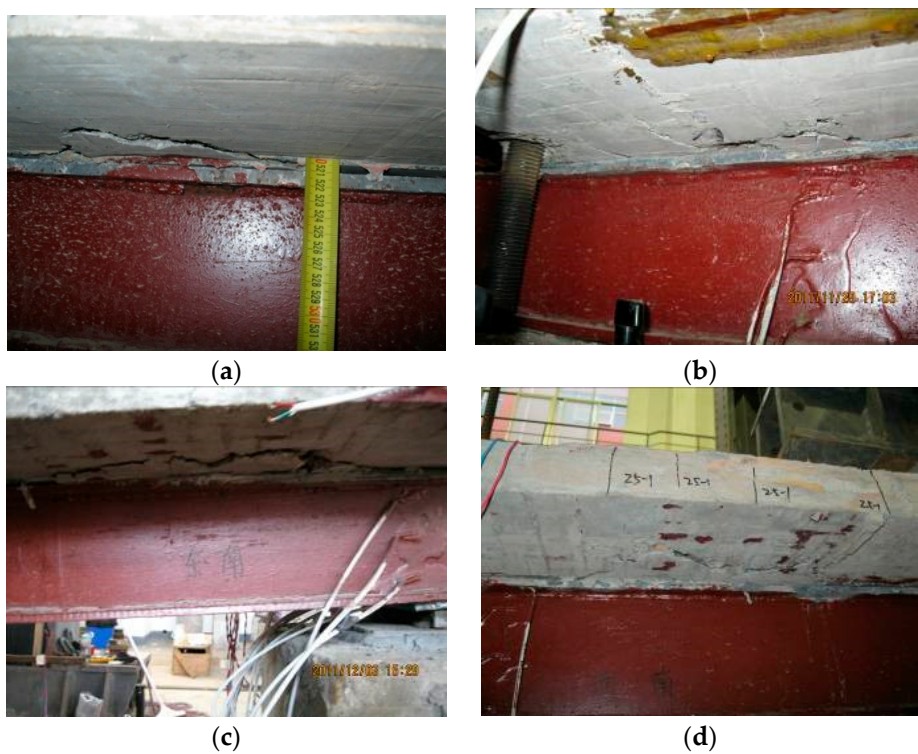

**Figure 7.** Failure mode of the specimens: (**a**) SCB-1; (**b**) SCB-2; (**c**) SCB-3; (**d**) SCB-4.

### 3.3. Bending Moment–Curvature Skeleton Curve

The bending moment–curvature skeleton curve of the four composite box girder specimens is shown in Figure 9. The skeleton curves of the specimen under the action of a low cycle reciprocating load have experienced three stages: approximate elasticity, elastoplastic, and failure stage. Before cracking, the skeleton curve of the specimen is approximately straight; after cracking, the skeleton curve of the specimen begins to bend. After the load exceeded the limit load, a certain approximate platform segment appeared on the skeleton curve of the specimen, and then the stiffness of the specimen was further reduced and finally destroyed. The skeleton curves of SCB–2 and SCB–4 are relatively full, and positive and negative bearing capacity are more than 13% larger than that of SCB–1. It shows that under the same conditions, the increase in the degree of shear connection can improve the seismic performance of the composite box girder and the ultimate bearing capacity. The ultimate bearing capacity of SCB–4 is nearly 30% higher than that of SCB–2, indicating that the width–thickness ratio of the web is greatly affected by the bearing capacity of the composite box girder, and the ultimate bearing capacity of the steel–concrete composite box girders can be greatly improved by improving the width–thickness ratio

of the web. The reason for the low level of SCB–3 carrying capacity relative to the test piece SCB–4 is that the web is too thin, resulting in local buckling of the composite box girder, which reduces the overall performance of the structure. The load–deflection skeleton curves of the four composite box beams have a long approximate platform segment, and the negative direction is shorter, indicating that the positive deformation capacity of the composite box girder is much stronger than that of the negative deformation.

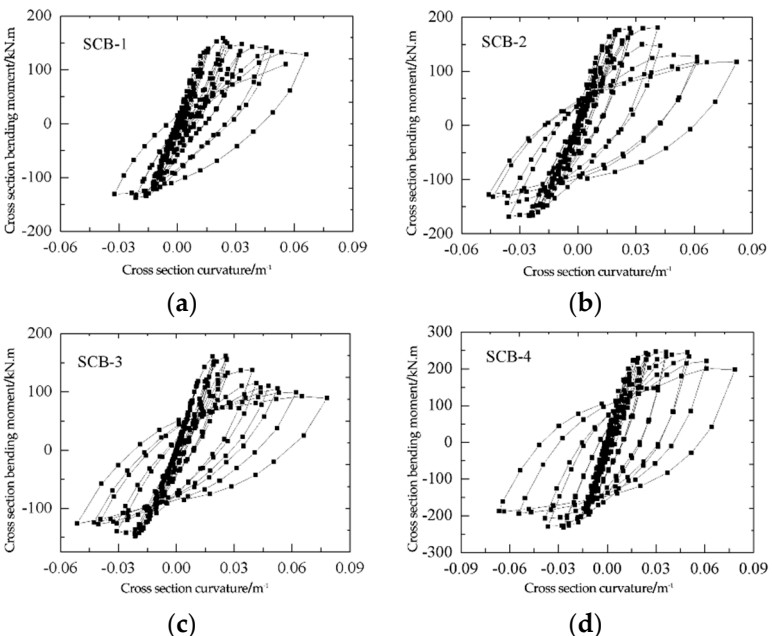

**Figure 8.** Bending moment–curvature hysteretic curves of mid-span: (**a**) SCB–1; (**b**) SCB–2; (**c**) SCB–3; (**d**) SCB–4.

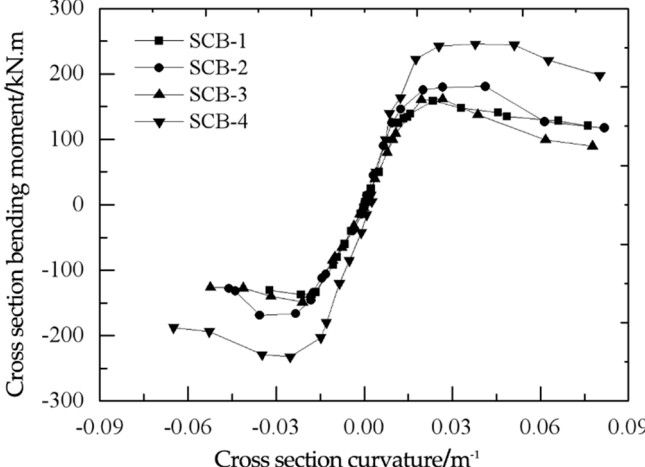

**Figure 9.** Bending moment–curvature envelope curves of mid-span.

## 4. Bending Moment–Curvature Restoring Force Model

### 4.1. The Basic Assumptions

1.  The concrete slab of the tensile zone is not involved in the work;
2.  The curvature of the steel girder and the concrete slab in the elastic stage is the same as that of the flat section;
3.  The elastic stiffness of the steel–concrete composite box girder with a partial shear connection is between the elastic stiffness in the case of a complete shear connection and the elastic stiffness in the case of complete non-connection, and assumes that the

elastic stiffness can be interpolated by the power function of the shear force in the case of partial shear connections [25].

*4.2. Combination Coefficient*

The combination coefficient of a positive bending moment and negative bending moment can be obtained according to the references [25,26]:

1. Positive bending moment

$$(EI)_e = (E_s I_s + E_c I_c)(1 + \psi_1) \tag{1}$$

$$\psi_1 = \frac{E_c A_c (d - 0.5 h_c) d_c}{(E_s I_s + E_c I_c)} \tag{2}$$

$$d = 0.5 h_c + \frac{E_s A_s d_c}{(E_s A_s + E_c A_c)} \tag{3}$$

Therefore, the combined coefficient of the full shear connection under the positive bending moment is $\psi_1$. In the formula: $E_s$ is the elastic modulus of steel; $I_s$ is the moment of inertia of the steel girder section; $I_c$ is the moment of inertia of the concrete slab section; $A_s$ is the section area of the steel beam; $A_c$ is the section area of the concrete slab; $h$ is the section height of the composite beams; $h_c$ is the cross section height of the concrete slab; $d_c$ is the distance between the steel girder centroid point and the concrete slab centroid point; $h_{s2}$ is the distance between the steel girder and the steel beam floor; $(EI)_e$ is the equivalent flexural rigidity of the composite beams.

2. Negative bending moment

$$(EI)'_e = E_s I_s (1 + \psi_2) \tag{4}$$

$$\psi_2 = \frac{E_r A_r d d_r}{(E_s I_s)} \tag{5}$$

$$d = \frac{E_s A_s d_r}{(E_r A_r + E_s A_s)} \tag{6}$$

Therefore, the combined coefficient of the total shear connection under the negative bending moment is $\psi_2$.

In the formula: $E_r$ is the elastic modulus of reinforcement; $A_r$ is the sectional area of reinforcement; $h_r$ is the distance between the interfacial interface of reinforcement; $d_r$ is the distance between the steel bar center and the girder center.

Therefore, the combination coefficient $\psi$ of the arbitrary shear connection degree can be obtained [10,26,27]:

$$\psi_1 = \frac{r^{0.5} E_c A_c (d - 0.5 h_c) d_c}{(E_s I_s + E_c I_c)} \tag{7}$$

$$\psi_2 = \frac{r'^{0.5} E_r A_r d d_r}{(E_s I_s)} \tag{8}$$

$$r = \frac{n}{n_f} \tag{9}$$

$$N_v = 0.43 A_{st} \sqrt{E_c f_c} \le \alpha A_{st} f_u \tag{10}$$

In the formula: $N_v$ is the shear bearing capacity of a single stud; $n_f$ is the total number of studs required for full shear connection, $n_f = \frac{V}{N_v}$; $n$ is the total number of studs; $V$ is the longitudinal shear force on the interface between the steel girder and concrete slab. The calculation method of the V value in the positive bending moment zone and the negative moment zone is different, and the specific calculation method can be calculated according to DL/T 5085-1999 [23]; $f_c$ is the axial compressive strength of concrete; $f_u$ is the ultimate

tensile strength of the stud; $A_{st}$ is the maximum sectional area of the stud; $f_{cu}$ is the cube compressive strength of the concrete. The value of parameters in DL/T 5085-1999 [23] are conservative, and Nie J.G., etc. [28], revised it on the basis of a lot of testing, and the values are as follows:

$$\alpha = \begin{cases} 0.7 & f_{cu} \leq 40\,\text{Mpa} \\ 0.7 + 0.014(f_{cu} - 40) & 40 < f_{cu} \leq 50\,\text{Mpa} \\ 0.84 & f_{cu} > 50\,\text{Mpa} \end{cases} \tag{11}$$

*4.3. Yield Moment*

The sectional strain diagram of composite beams with shear connections shown in Figure 10 is known as follows:

1. Positive bending moment

$$M = \varphi(EI)_e: \tag{12}$$

$$(EI)_e = E_s I_s + E_c I_c + E_s A_s (h_{s1} - e) d_c \tag{13}$$

$$e = h_{s1} - \frac{\psi_1 (E_s I_s + E_c I_c)}{(E_s A_s d_c)} \tag{14}$$

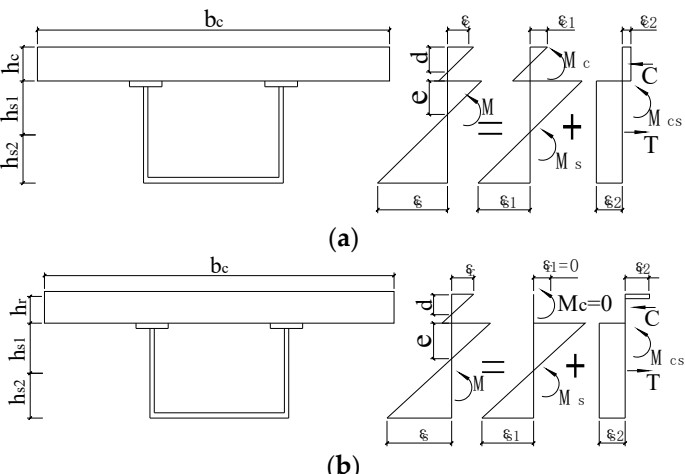

**Figure 10.** Section strain of composite box girder with part shear connection: (**a**) positive bending moment; (**b**) negative bending moment.

The following formula is satisfied when the steel girder bottom plate yields:

$$\varphi = \frac{f_y}{[E_s(h_s - e)]} \tag{15}$$

$$M_y = (EI)_e \frac{f_y}{[E_s(h_s - e)]} \tag{16}$$

In the formula: $f_y$ is the yield strength of the steel beam bottom plate.

2. Negative bending moment

$$(EI)'_e = E_s I_s + E_s A_s (h_{s1} - e) d_r \tag{17}$$

$$e = h_{s1} - \frac{I_s \psi_2}{(A_s d_r)} \tag{18}$$

The following formula is satisfied when the steel girder bottom plate yields:

$$\varphi = \frac{f_y}{[E_s(h_s - e)]} \tag{19}$$

$$M'_y = (EI)'_e \frac{f_y}{[E_s(h_s - e)]} \tag{20}$$

### 4.4. Composite Box Girder Bending Moment–Curvature Restoring Force Model

The restoring force model consists of two parts: the skeleton curve and hysteretic rule. The method of determining the recovery force model is mainly the experimental fitting method, system identification method, and theoretical calculation method [29]. At present, the elastic–plastic restoring force model is divided into curve type and fold line type, and the fold line type model is widely adopted because of its simple application. Currently, the model of the fold line restoring force is mainly composed of two lines, three lines, four lines, fixed-point orientation, maximum point orientation, slip type, slip mixed type, degenerate double line, and degenerate trilinear, etc.

### 4.4.1. Bending Moment–Curvature Skeleton Curve Model

The bending moment–curvature recovery model of the composite box girder can be determined according to the test. In this paper, the skeleton curves of the composite box girder under the action of low cyclic reciprocating load have undergone three stages: approximate elasticity, elastoplastic, and failure stage. Therefore, the composite box girder bending moment–curvature model can be approximated by the degenerate three-fold model, as shown in Figure 11.

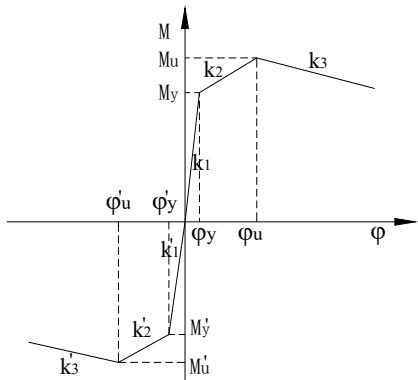

**Figure 11.** M-φ envelope curves model.

Determining the composite box girder bending moment–curvature skeleton curve model needs 10 key parameters: positive yield bending moment $M_y$, positive elastic stiffness $k_1$, positive plastic limit bending moment $M_u$, positive reinforcement stiffness $k_2$, positive descending slope stiffness $k_3$, negative elastic stiffness $k'_1$, negative yield bending moment $M_y$, negative plastic limit bending moment $M'_u$, negative reinforcement stiffness $k'_2$, and negative descending slope stiffness $k'_3$.

3.    Positive elastic stiffness $k_1$

$$k_1 = (E_s I_s + E_c I_c)(1 + \psi_1) \tag{21}$$

$$d = 0.5h_c + \frac{E_s A_s d_c}{(E_s A_s + E_c A_c)} \tag{22}$$

4.    Positive yield bending moment $M_y$

$$M_y = k_1 \frac{f_y}{[E_s(h_s - e)]} \tag{23}$$

$$e = h_{s1} - \frac{\psi_1(E_s I_s + E_c I_c)}{(E_s A_s d_c)} \tag{24}$$

5. Positive plastic limit bending moment $M_u$ [30]

$$M_u = M_{su} + r^{0.5}\left(M_{fu} - M_{su}\right) \tag{25}$$

In the formula: $M_{fu}$ is the plastic ultimate bearing capacity of a composite beam with full shear connection, and the computational method is according to reference [23]; $M_{su}$ is the plastic ultimate bearing capacity of the steel girder.

6. Positive reinforcement stiffness $k_2$

$$k_2 = \beta_1 k_1 \tag{26}$$

In the formula: $\beta_1$ is the reduction factor of strengthening stiffness, and according to the statistical analysis of the experimental data and the literature [31], $\beta_1 = 0.314r^{-1}\left(\frac{h_c}{h_s}\right)^{1.5}$.

7. Positive descending slope stiffness $k_3$

$$k_3 = \beta_2 k_1 \tag{27}$$

In the formula: $\beta_2$ is the reduction factor of stiffness in the descending section, and according to the statistical analysis of test data in this paper, $\beta_2 = 6.65r^{1.5}(h_c/h_s)^{3.5}$.

8. Negative elastic stiffness $k_1'$c

$$k_1' = E_s I_s(1 + \psi_2) \tag{28}$$

$$\psi_2 = \frac{r'^{0.5} E_r A_r d d_r}{(E_s I_s)} \tag{29}$$

$$d = \frac{E_s A_s d_r}{(E_r A_r + E_s A_s)} \tag{30}$$

9. Negative yield bending moment $M_y'$

$$M_y' = k_1' \frac{f_y}{[E_s(h_s - e)]} \tag{31}$$

$$e = h_{s1} - \frac{I_s \psi_2}{(A_s d_r)} \tag{32}$$

10. Negative plastic limit bending moment $M_u'$

The calculation of $M_u'$ can be referred to the literature [32]: after the area of the longitudinal bar is reduced by the shear strength, the simplified plastic method is used to calculate. The reduction formula for the area of the longitudinal bar is $A_r' = \frac{nN_v}{f_r}$, and $f_r$ is the yield strength of the reinforcing bar.

11. Negative reinforcement stiffness $k_2'$

$$k_2' = \beta_1' k_1' \tag{33}$$

In the formula: $\beta_1'$ is the negative reduction factor of stiffness in the descending section, and according to the statistical analysis of test data in this paper, $\beta_1' = 1.67\beta_1$.

(10) Negative descending slope stiffness $k_3'$

$$k_3' = \beta_2' k_1' \tag{34}$$

In the formula: $\beta'_2$ is the negative reduction factor of stiffness in the descending section, and according to the statistical analysis of test data in this paper, $\beta'_2 = 2\beta_2$.

### 4.4.2. Model Verification of Bending Moment–Curvature Skeleton Curve

In this paper, four composite box girder specimens were calculated using the steel–concrete composite box girder skeleton curve model and were compared with the experimental results, and the comparison results are shown in Figure 12. It can be seen that the skeleton curve model proposed in this paper is in good agreement with the four specimen skeleton curves. It is proved that the skeleton curve proposed in this paper is reasonable, and the accuracy of the bending moment–curvature elastic stiffness and bending moment calculation method proposed in Section 2 is verified. The calculation of bending moment–curvature skeleton model of the steel–concrete composite box girder is simple, convenient for manual calculation, and suitable for engineering applications.

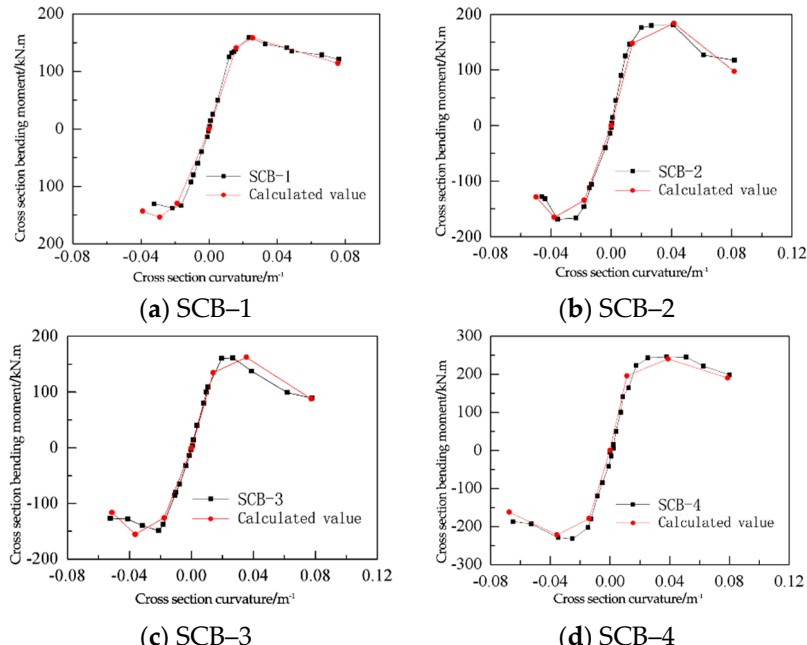

**Figure 12.** Comparison of bending moment–curvature skeleton model and test skeleton curve.

### 4.4.3. Bending Moment–Curvature Hysteresis Model and Its Verification

The key points for determining the hysteretic rule are the positive and negative unloading stiffness and the reloading route. The hysteresis curve of the test shows that the unloading stiffness of the specimens has little change before the load reaches the yield load. After the load reaches the yield load, the unloading stiffness decreases with the increase of the plastic displacement. The reloading routes basically point to the highest point of the previous cycle. Therefore, the composite box girder bending moment–curvature restoring force model can be simplified as the maximum point pointing model of the degenerate trifold line, as shown in Figure 13.

Each fold point number represents the hysteresis rule walking routes, starting from zero to the end of the 26 points according to the numerical sequence. When the load is less than the yield bending moment, initial stiffness $k_1$ and $k'_1$ are taken, respectively, by positive and reverse loading stiffness; after the load is greater than the yield load, $k_4$ and $k'_4$ are, respectively, taken from the positive and negative unloading stiffness and unloaded to zero. The forward and backward loading routes basically point to the maximum point of positive and negative loading. If the forward and backward loading values are unloaded before reaching the skeleton curve, the forward and backward stiffness are, respectively, taken as $k_4$ and $k'_4$. If it is not unloaded to zero and then reloaded, the continued loading curve points to the intersection of the extension line of the previous loading curve and the

skeleton curve, such as lines 13–14–15 and 21–22–23 in Figure 13. When the first reverse loading occurs, the load path points to the yield point on the skeleton curve, as shown in Figure 13, line 3–4.

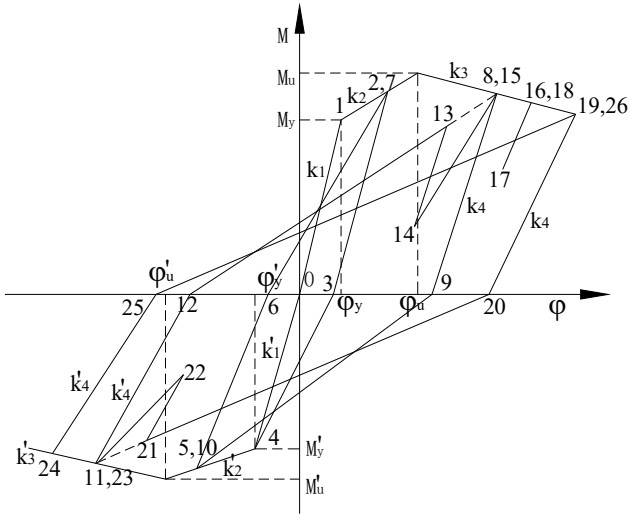

**Figure 13.** Bending moment–curvature restoring force model under cyclic load.

The positive and negative unloading stiffness and the vertex of each hysteresis ring of four specimens were calculated, and the relationship between the stiffness and the section curvature at the vertex of the corresponding hysteresis ring was described in Figure 14, and using ORIGIN software on the exponential function fitting, the fitting results are shown in Figure 14.

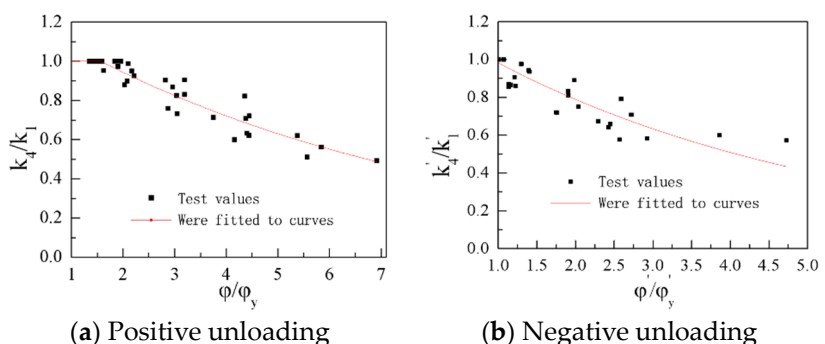

(**a**) Positive unloading  (**b**) Negative unloading

**Figure 14.** Fit of stiffness deterioration.

The positive unloading stiffness fitting formula:

$$k_4 = \beta_3 k_1 \tag{35}$$

$$\beta_3 = \begin{cases} 1 & \frac{\varphi_r}{\varphi_y} < 1.6 \\ 1.23 e^{\left(\frac{-0.134\varphi_r}{\varphi_y}\right)} & \frac{\varphi_r}{\varphi_y} \geq 1.6 \end{cases} \tag{36}$$

The negative unloading stiffness fitting formula:

$$k_4' = \beta_3' k_1' \tag{37}$$

$$\beta_3' = \begin{cases} 1 & \frac{\varphi_r'}{\varphi_y'} < 1 \\ 1.22 e^{\left(\frac{-0.22\varphi_r'}{\varphi_y'}\right)} & \frac{\varphi_r'}{\varphi_y'} \geq 1 \end{cases} \tag{38}$$

In the formula: $\varphi_r$ and $\varphi_r'$ represent the positive and negative curvature maximums that have been achieved after yield.

The recovery force model of the hysteretic curve proposed in this paper was used to calculate the sample SCB-1~SCB-4 and compare with the experimental results, as shown in Figure 15. It can be seen that the calculation curve of the recovery force model proposed in this paper is in good agreement with the test curve, and the accuracy of the model is verified.

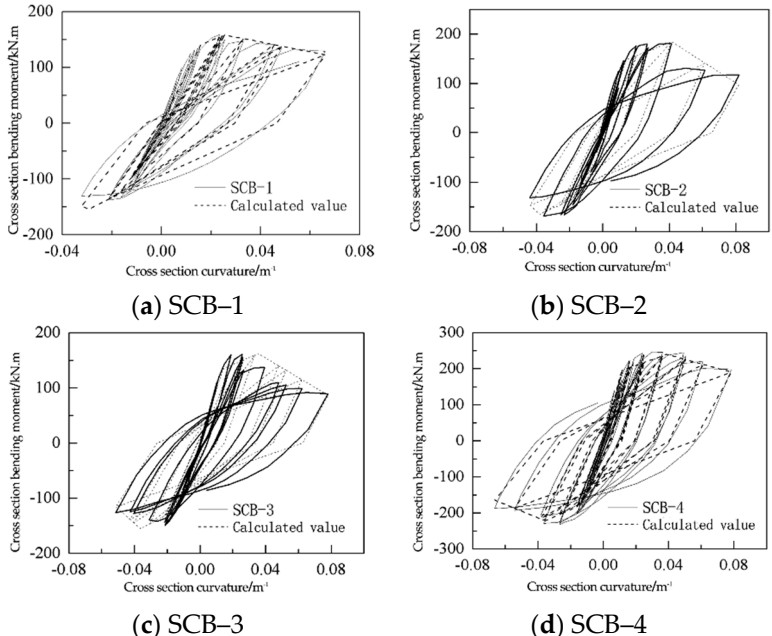

(**a**) SCB–1

(**b**) SCB–2

(**c**) SCB–3

(**d**) SCB–4

**Figure 15.** Comparison of bending moment–curvature restoring force calculation model and test curve.

## 5. Conclusions

Steel and concrete composite box girder cross section bending moment–curvature hysteretic characteristics are one of the most important features for the composite box girder. The establishment of the bending moment–curvature hysteresis rule of steel–concrete composite box girders provides a foundation for the study of the restoring force model of composite box girders and even composite frame structures, and provides a theoretical model for the elastic–plastic analysis of composite box girder structures.

(1) Composite box girder moment–curvature hysteresis curves can be divided into three stages: elasticity stage, elastoplastic stage, and failure stage. The load–deflection hysteresis rings of composite box girders with different shear connection degrees and height–thickness ratios are plump, and there is no obvious pinching phenomenon, and have good seismic performance.

(2) The skeleton curves of the composite box girder underwent three stages: approximate elasticity stage, elastoplastic stage, and failure stage. Under the same conditions, the frame curve of the composite box girder with a large shear connection is fuller and the seismic performance is better. The skeleton curves have long approximate platform segments in the positive direction and short approximate platform segments in the negative direction, indicating that the positive deformation capacity of the composite box girder is much stronger than the negative deformation capacity, and the positive ductility ratio of the composite box girder is much greater than the negative ductility ratio. The positive and negative ductility ratios of composite box girders increase obviously with the increase of height–thickness ratio.

(3) The influence of the shear connection degree on the bending stiffness of composite box girders is considered by using the power function interpolation method, and the

calculation method of forward and negative cross section bending moment–curvature elastic stiffness of the composite box girder considering interface slips is put forward. Moreover, the expression of the bending moment of the section yield is obtained, and the accuracy of the method is verified by comparison with the test results. A three-fold model of the bending moment–curvature skeleton curve of steel–concrete composite box girder sections is established and compared with the experimental results in this paper. The calculation model is in good agreement with the experimental structure, and the calculation method of the model is simple and convenient for engineering applications.

(4) On the basis of experimental data and theoretical analysis, the expressions of the positive and negative stiffness degradation of composite box girders were put forward, and the pointing hysteretic model of the bending moment–curvature degradation vertex of steel–concrete composite box girders was established. The calculated curve of the model was in good agreement with the experimental curve, and the calculation method of the model was simple and clear, convenient for hand calculation, and suitable for engineering applications.

**Author Contributions:** J.Q.: Conceptualization and methodology; Y.Y.: Validation and Formal analysis; Z.H.: Writing—Original Draft; W.L.: Writing—Review & Editing; W.Z.: Visualization; F.L.: Resources; J.W.: Data Curation. All authors have read and agreed to the published version of the manuscript.

**Funding:** The research described in this paper was supported by Department of Education of Hunan Province: the Research Foundation of Education Bureau of Hunan Province, China (Grant No. 20A184), Department of Science and Technology of Hunan Province: the Natural Science Foundation of Hunan Province, China (Grant No. 2021JJ30261), Department of Education of Hunan Province: the postgraduate scientific research innovation project of Hunan Province, China (Grant No. CX20210993).

**Institutional Review Board Statement:** Not applicable.

**Informed Consent Statement:** Not applicable.

**Data Availability Statement:** Our data is publicly available, experimentally obtained. The authors will supply the relevant data in response to reasonable requests.

**Conflicts of Interest:** The authors declare no conflict of interest.

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
