# Peer review of "Experimental Study and Theoretical Analysis of Steel–Concrete Composite Box Girder Bending Moment–Curvature Restoring Force"

_sustainability, doi:10.3390/su15086585_

Round 1

Reviewer 1 Report

Figure 14 is not clear;

The failure mode is not presented;

why didnt present the strain results?

Author Response

  Thank you for your valuable comments on our article. All your comments are both professional and valuable, which are great significance to improve my scientific research value and the quality of essay writing. The following are my response to your comments one by one.

Point 1: Figure 14 is not clear

Response 1: We’ve replaced it with a clearer Figure.

Point 2: The failure mode is not presented

Response 2: We have added the failure mode on page 7.

Point 3: Why didn’t present the strain results?

Response 3: Because the results of strain have little relation to the restoring force model, it is not shown in this paper due to the length of the paper.

Reviewer 2 Report

Title: Experimental study and theoretical analysis of steel-concrete

composite box girder bending moment-curvature restoring force

In this paper the low cycle vertical load tests of steel-concrete composite box girders were conducted with different shear connection degree and ratio of web height to thickness have been investigated.

This paper can be interesting but some questions need to be clarified before a possible publication.

- This type of structural components has been extensively studied in the past, the Authors have to clarify better the novelties of this work.

- Section 2: all the studied speciments (four) are different (see Table 1) how can the statistic consideration be done with the experimental results obtained?

- It may be interesting to see the girders after testing and the failure zones.

- Can occur buckling problems when steel box is compressed?

- How was o equation 10) obtained?

Author Response

Thank you for your valuable comments on our article. All your comments are both professional and valuable, which are great significance to improve my scientific research value and the quality of essay writing. The following are my response to your comments one by one.

Point 1: This type of structural components has been extensively studied in the past, the Authors have to clarify better the novelties of this work

Response 1: In previous studies, the restoring force model of steel-concrete composite box girder was relatively rare, and the influence of shear connection degree and web height to thickness ratio on the restoring force model was not considered. And the restoring force model is based on component, which is not easy to be applied in engineering. In this paper, a restoring force model of steel-concrete composite box girder considering the effects of shear section and web height to thickness ratio is established. The calculation method of the model is simple and easy to calculate by hand. We’ve added relevant explanations about the innovation of this study in the introduction.

Point 2: Section 2: all the studied specimens (four) are different (see Table 1) how can the statistic consideration be done with the experimental results obtained?

Response 2: The four specimens of this test mainly consider the change of shear connection degree, and analyze the influence of shear connection degree on the restoring force model of composite box girder. Existing statistical research has been done on the bending moment-curvature elastic stiffness of steel-concrete composite box girder with partial shear connection. According to literature 28, 30-31, the moment-curvature elastic stiffness of steel-concrete composite box girder with partial shear connection is between the elastic stiffness in the case of complete shear connection and the elastic stiffness in the case of complete shear non-connection, and it is assumed that the elastic stiffness in the case of partial shear connection can be interpolated according to the power function of shear connection.

Point 3: It may be interesting to see the girders after testing and the failure zones.

Response 3: On page 6, we have added the relevant contents and pictures of the failure modes of composite beams under low cycle reciprocating loads.

Point 4: Can occur buckling problems when steel box is compressed?

Response 4: In fact, we have added a diaphragm every 500mm to the web of composite beams to prevent local buckling of the web. However, specimen SCB-3 still showed local buckling of failure of the web as shown in Figure 7 on page 7.

Point 5: How was o equation 10) obtained?

Response 5:Equation 10) is derived from the research content in Reference [27].

Reviewer 3 Report

1- The abstract missed mentioning any motivation for this research and any quantitative results, please revise it 

2- The authors did not mention the novelty of the work compared with other studies. I think there is a lot of work available related to the subject. The introduction lakes how and why this work is novel and what makes it different. It would be worth analyzing the results with an interpretation where the gap of the study in the field.

3- The motivation of this paper is missing. What is the problem that author(s) would like to solve? The overall introduction section is incomplete and difficult to understand the reason for conducting this investigation. Put more effort into explaining in depth why this study is critical for China and the world.

4- The authors need to explain the reason behind selecting the parameters and their values/range in Table 1

5- Why did you select different strengths (yield and ultimate) for each steel plate with different thicknesses in Table 2? Same thing for the reinforcement bars

6- Use a better version of Figure 3 and Figure 4

7- Since the authors are examining the performance under a seismic or repeated loading, it is required to calculate and present the effect of different configurations and parameters on the energy dissipation and viscous damping 

Author Response

Thank you for your valuable comments on our article. All your comments are both professional and valuable, which are great significance to improve my scientific research value and the quality of essay writing. The following are my response to your comments one by one.

Point 1: The abstract missed mentioning any motivation for this research and any quantitative results, please revise it.

Response 1: We’ve already revised it.

Point 2: The authors did not mention the novelty of the work compared with other studies. I think there is a lot of work available related to the subject. The introduction lakes how and why this work is novel and what makes it different. It would be worth analyzing the results with an interpretation where the gap of the study in the field.

Response 2: In previous studies, the restoring force model of steel-concrete composite box girder was relatively rare, and the influence of shear connection degree and web height to thickness ratio on the restoring force model was not considered. And the restoring force model is based on component, which is not easy to be applied in engineering. In this paper, a restoring force model of steel-concrete composite box girder considering the effects of shear section and web height to thickness ratio is established. The calculation method of the model is simple and easy to calculate by hand. We’ve already revised it in the introduction.

Point 3: The motivation of this paper is missing. What is the problem that author(s) would like to solve? The overall introduction section is incomplete and difficult to understand the reason for conducting this investigation. Put more effort into explaining in depth why this study is critical for China and the world.

Response 3: The seismic code stipulates that it is necessary to perform elastoplastic analysis of structures when they are subjected to rare earthquakes. In order to accurately study the elastic-plastic seismic response of the structure, the restoring force model of the building structure is the primary problem that needs to be solved. However, the current research on the seismic performance of composite beam mainly focuses on the composite I-beam, and the research on the resistance theory of composite box girder members lags behind the practical engineering application of composite box girder. It is of great significance to carry out seismic research on composite box girder members, rationally analyze and evaluate the seismic performance of composite box girder, to improve the theory and method of seismic design of composite box girder, to improve the ability of major engineering structures to resist earthquake disasters, and to promote the application of composite box girder in mega-structures and long-span structures. We’ve already revised it in the introduction.

Point 4: The authors need to explain the reason behind selecting the parameters and their values/range in Table 1.

Response 4: According to the existing research, the two parameters of shear connection degree and web height to thickness ratio have great influence on the seismic performance of composite box girder. On the one hand, the different degree of shear connection will lead to different degrees of slippage at the interface, which directly affects the stiffness of the composite box girder. On the other hand , the ratio of height to thickness of web will affect the seismic bearing capacity of composite box girder. Therefore, these test specimens mainly considered the changes of these two parameters. The values of each parameter are obtained by referring to the recommended values commonly used in engineering practice, and then adjusting according to the test conditions and actual material properties .

Point 5: Why did you select different strengths (yield and ultimate) for each steel plate with different thicknesses in Table 2? Same thing for the reinforcement bars

Response 5: Because steel and steel bars not purchased from the same batch and the same manufacturer, there are certain differences of the yield strength and ultimate strength in the actual material test.

Point 6: Use a better version of Figure 3 and Figure 4.

Response 6: We’ve already revised it.

Point 7: Since the authors are examining the performance under a seismic or repeated loading, it is required to calculate and present the effect of different configurations and parameters on the energy dissipation and viscous damping.

Response 7: In fact, we have studied the energy dissipation capacity, ductility, deformation recovery capacity and stiffness degradation of composite box girder, but due to the length limitation of the paper, it is not shown in this paper.

Reviewer 4 Report

The paper presents “Experimental study and theoretical analysis of steel-concrete composite box girder bending moment-curvature restoring force.” After deep evaluation, the paper is interesting and well-written. The manuscript explains the methodology well, has clear results, and has a good simulation for specimens. This study provides a significant finding concerning construction. Therefore, I recommend accepting the paper for publication.

Author Response

Thank you for your time and effort in reviewing the manuscript, and thank you for your recognition of our work and the content of the article

Round 2

Reviewer 2 Report

Accept in present form

Reviewer 3 Report

The authors conduction the required revision